# Expression Patterns of the Heat Shock Protein 90 (Hsp90) Gene Suggest Its Possible Involvement in Maintaining the Dormancy of Dinoflagellate Resting Cysts

**DOI:** 10.3390/ijms222011054

**Published:** 2021-10-13

**Authors:** Yunyan Deng, Fengting Li, Zhangxi Hu, Caixia Yue, Ying Zhong Tang

**Affiliations:** 1CAS Key Laboratory of Marine Ecology and Environmental Sciences, Institute of Oceanology, Chinese Academy of Sciences, Qingdao 266071, China; yunyandeng@qdio.ac.cn (Y.D.); lifengting@qdio.ac.cn (F.L.); zhu@qdio.ac.cn (Z.H.); yuecaixia@qdio.ac.cn (C.Y.); 2Laboratory of Marine Ecology and Environmental Science, Qingdao National Laboratory for Marine Science and Technology, Qingdao 266237, China; 3Center for Ocean Mega-Science, Chinese Academy of Sciences, Qingdao 266071, China; 4University of Chinese Academy of Sciences, Beijing 100049, China

**Keywords:** dinoflagellate, dormancy, environmental cDNA library, heat shock protein 90 (Hsp90), resting cysts, *Scrippsiella trochoidea*, temperature stress

## Abstract

Heat shock protein 90 (Hsp90) is a highly conserved molecular chaperone functioning in cellular structural folding and conformational integrity maintenance and thus plays vital roles in a variety of biological processes. However, many aspects of these functions and processes remain to be fully elucidated, particularly for non-model organisms. Dinoflagellates are a group of eukaryotes that are exceedingly important in primary production and are responsible for the most harmful algal blooms (HABs) in aquatic ecosystems. The success of dinoflagellates in dominating the plankton community is undoubtedly pertinent to their remarkable adaptive strategies, characteristic of resting cyst production and broad tolerance to stresses of temperature and others. Therefore, this study was conducted to examine the putative roles of *Hsp90* in the acclimation to temperature stress and life stage alterations of dinoflagellates. Firstly, we isolated the full-length cDNA of an Hsp90 gene (*StHsp90*) via RACE from the cosmopolitan HAB species *Scrippsiella trochoidea* and tracked its transcriptions in response to varied scenarios via real-time qPCR. The results indicated that *StHsp90* displayed significant mRNA augment patterns, escalating during 180-min treatments, when the cells were exposed to elevated and lowered temperatures. Secondly, we observed prominently elevated *St**Hsp90* transcriptions in the cysts that were stored at the cold and dark conditions compared to those in newly formed resting cysts and vegetative cells. Finally, and perhaps most importantly, we identified 29 entries of *Hsp90*-encoding genes with complete coding regions from a dinoflagellate-specific environmental cDNA library generated from marine sediment assemblages. The observed active transcription of these genes in sediment-buried resting cysts was fully supported by the qPCR results for the cold-stored resting cysts of *S. trochoidea*. *Hsp90s* expressions in both laboratory-raised and field-collected cysts collectively highlighted the possible involvement and engagement of Hsp90 chaperones in the resting stage persistence of dinoflagellates.

## 1. Introduction

Molecular chaperones fulfill foundational and vital functions in cellular proteostasis, including facilitating the proper folding of native proteins and/or stabilizing and refolding of misfolded ones, thus preventing aberrant aggregation [1,2,3]. Heat shock protein 90 (Hsp90) is one of the most conserved and abundant molecular chaperones, exists in all species except for archaea, and accounts for ~1% of the whole cellular cytosolic proteins [4]. Commonly, in a wide variety of organisms, like many other Hsp chaperones being highly stress-inducible, Hsp90s participate in defense responses against a broad range of environmental cues that could potentially damage the cellular and molecular structures, such as thermal shock, heavy metal exposure, nutrient limitation, oxidative damage, and light/dark transition [1,5,6]. The roles of Hsp90s, however, extend well beyond stress acclimation. In contrast to other intensively studied Hsp chaperones, such as Hsp70s, the 90-kDa Hsps are not necessary for the *de novo* folding of native proteins but, rather, assist in their final maturation, which is considered to complement an additional layer of regulation in some cases [4,7,8]. Distinguished from Hsp70s in having a broad substrate range, Hsp90s seem to have a select set of clients [9]. The known clients of Hsp90 members to date are enriched in regulatory proteins, such as steroid hormone receptors, transcription factors, kinases, and signal transduction proteins [4,9,10]. These clients reside in nearly every developmental and signaling pathway in eukaryotes, providing a plausible explanation for the broad influences of Hsp90s on the relationship between the genotype and phenotype and pertaining to diverse and even seemingly unrelated physiological processes (e.g., homeostasis maintenance, signal transduction, cell cycle control, and protein trafficking) [2,3,4,6,10,11].

Dinoflagellates (division Pyrrhophyta, class Dinophyceae) consist of a diverse and ubiquitous group of unicellular protists that occupy vital niches in marine and freshwater habitats. As the second-most abundant phytoplankton in a coastal marine ecosystem, they are notable for participating in ocean primary productivity and food webs and, thus, contribute to carbon fixation and O_2_ production [12]. Many photosynthetic members in this lineage receive great attention for their symbiosis with corals and other invertebrates, which is essential to the coral reef ecosystem and the biodiversity paradise in the world’s oceans [13]. Members in this group are also infamous for being the greatest agents of harmful algal blooms (HABs), since ~40% of the recorded HAB-forming species are dinoflagellates [14]. Dinoflagellate HABs detrimentally affect commercial and recreational fisheries, aquaculture, and coastal tourism, as well as human and wildlife health. Mortality or physiological impairments either due to toxin production or oxygen depletion and physical clotting are associated with a massive accumulation of biomass blooms [15]. The ecological success of dinoflagellates in dominating the plankton community is undoubtedly pertinent to their remarkable adaptive strategies, such as resting cyst production and a broad tolerance to stressors like thermal and cold shock, which is reflected by a tremendous diversity in the form and nutrition and an extensive fossil record dating back several hundred million years ago [12,16]. Temperature is a vital environmental factor that influences a broad range of physiological processes and, thus, limits the biomass abundance and geographical distribution [17]. Some HAB-forming dinoflagellate species are eurythermal organisms whose vegetative cells show amazing resilience to temperature fluctuations and, thus, facilitate to gain broader niches relative to their competitors and grazers in ecosystems [18,19,20,21]. A broad temperature tolerance has been demonstrated as an adaptive mechanism contributing to a wide distribution, population survival, and bloom formation of these species [20,21]. Many phytoplankton species include a resting stage in their life histories, which offers an adaptive advantage to endure harsh environmental extremes that are unsuitable for the growth and/or survival of vegetative cells [22]. The resting stage of dinoflagellates, also well-known as resting cysts [23], have been extensively investigated due to their roles in the biology and ecology of dinoflagellates, as they are involved in the resistance to a variety of unfavorable environments, protection from grazers or parasite attacks, bloom dynamics and recurrence, and geographic expansion of populations [23,24,25]. Heretofore, the molecular basis underlying the adapting (or surviving) abilities of dinoflagellates under adverse conditions, such as the functional genes and their precise expressions involved in these processes, however, have been rarely investigated and, therefore, should be further explored.

Although the essential cellular functions of Hsp90s have been intensively investigated in mammalian and fungal models and amassed data has continued to emphasize the importance of Hsp90 chaperones in terrestrial plant development and response to external insults, information concerning Hsp90 members in dinoflagellates is scarce, and the relevant study lags greatly behind when compared to that of other taxa [2,6]. Thus far, about 130 entries relevant to *Hsp90* from more than 40 dinoflagellate species are available in the public GenBank databases (accessed on August 2021), though most of them were fragments obtained for phylogenetic studies (i.e., generally for gaining phylogenetic inference and/or species delineation [26,27,28,29,30,31]), while the rest were mainly yielded via EST (expressed sequence tags) and -omics studies or spliced leader sequence analyses [32,33,34,35,36,37,38,39,40]. The complete coding regions with expression patterns under environmental stimuli have been merely characterized in a few species of dinoflagellates [41,42,43,44]. For instance, thermally induced differential regulations of the *Hsp90*/*Hsp90* homologs were observed in *Symbiodinium* sp. [33,38,41], *Prorocentrum minimum* [42], and *P. donghaiense* [44]. Significantly elevated mRNA accumulations of *Hsp90* members were documented in *Oxyrrhis marina*, *P. minimum*, *P. donghaiense*, *Margalefidinium polykrikoides* (= *Cochlodinium polykrikoides*), and *Alexandrium pacificum* that were subjected to high salinity or chemical (copper, nitrogen, phosphorus, and NaOCl) discharges [34,40,42,43,44]. Besides, the Hsp90 protein abundance was reported to be remarkably enhanced at the S phase of the cell cycle of *P. donghaiense* and consistently maintains a higher level of toxicity loss in the mutant of *Alexandrium pacificum* (= *A. catenella*) than that in the wild-type strain [35,37], seemingly implying a pertinence of Hsp90 to cellular physiological processes in dinoflagellates. Nevertheless, little is known about the exact functions of Hsp90s in dinoflagellates, which highlights the necessity for a more decisive exploration on the functions of Hsp90s with well-designed experiments and examinations in this group of microalgae.

In the present study, a *Hsp90* gene (*StHsp90*) from the cosmopolitan and eurytherm (temperature tolerance range of 10–30 °C [18]) dinoflagellate *Scrippsiella trochoidea* was characterized with the aim to preliminarily examine the potential contribution of the gene to two well-documented adaptive (or survival) strategies for HAB-forming dinoflagellates, wide temperature acclimation, and resting cyst production. *S. trochoidea* was selected because it is a representative HAB species of dinoflagellates and readily produces resting cysts [45,46,47,48,49,50]. We then constructed a DinoSL (dinoflagellate mRNA-specific spliced leader [32])-based environmental cDNA (e-cDNA) library from the marine sediment and sequenced it with the third-generation sequencing platform. The Hsp90-encoding genes were identified from the dinoflagellate-specific full-length e-cDNA library, indicating an active expression of Hsp90s in the dinoflagellate resting cysts buried in marine sediment. Our results suggest a highly possible involvement and engagement of Hsp90 chaperones in the resting stage persistence and temperature stress adaptation of dinoflagellates. Our findings could make fundamental contributions towards a better understanding of the functions of Hsp members in this ecologically important microalgal lineage.

## 2. Results

### 2.1. The Full-Length cDNA Sequence of StHsp90

The products of 3′ and 5′ RACEs were 532 bp and 1210 bp in length, respectively. The full-length cDNA sequence of *StHsp90* was obtained by overlapping the two fragments with the 821-bp fragment amplified by primers P1 and P2. The newly generated sequence spanned 2474 bp, comprising a 106-bp 5’ UTR with the conserved dinoflagellate spliced leader (DinoSL) sequence, a 238-bp 3’ UTR ending in a poly (A) tail, and an ORF of 2130 bp (Appendix A). The obtained sequence was deposited in GenBank with the accession number MZ779085.

### 2.2. Structural Characterization of StHsp90

The *StHsp90* ORF had a 55.73% GC content and encoded a protein of 709 amino acid residues, accounting for a predicted molecular weight of 81.42 kDa and a theoretical isoelectric point of 4.96. For the predicted amino acid sequence, five characteristic signatures of the Hsp90 family at residues 25–46 (NKEIFLRELISNASDALDKIRY), 92–100 (LGTIAKSGT), 116–131 (IGQFGVGFYSAYLVAD), 340–349 (IKLYVRRVFI), and 366–380 (GVVDSEDLPLNISRE) were deduced by a BLAST analysis and compared with other Hsp90 counterparts (Appendix A). The conserved LXXLL sequence was detected at residues 640–644 (Appendix A). The cytosolic Hsp90-specific motif MEEVD was found at the C-terminus of the newly generated StHsp90 (Appendix A). The SignalP program analysis found that no signal peptide was present in the sequence. Four conformational states: *α*-helix, extended strand, *β*-turn, and random coil, were detected in the secondary structure, with the proportion of each type being 52.33%, 13.82%, 5.22%, and 28.63%, respectively, suggesting that *α*-helix was the major component of StHsp90.

### 2.3. Differential Transcriptional Responses of StHsp90 in Vegetative cells to Temperature Variations in Laboratory Culture

To examine the transcription pattern of *StHsp90* in response to temperature variations, vegetative cells kept at routine maintenance conditions (20 ± 1 °C) severed as the control group. Significant augments were detected in *StHsp90* transcriptions after being subjected to both the lower (i.e., from 20 °C to 5, 10, or 15 °C) and higher (i.e., from 20 °C to 25 or 30 °C) temperature stresses (ANOVA, *p* < 0.01; Figure 1A). The lowest expression was observed at 20 °C group, while the transcriptions increased with the magnitude of the temperature changes, along with both the decreasing and increasing directions. The mRNA levels for the group subjected to ±10 °C exposures were also tracked over time up to 180 min, both of which displayed a clear time-dependent response: for the 10 °C increase treatment, the relative *StHsp90* transcription was upregulated significantly after 10 min and reached the peak at 180 min, which was 16-fold that in the control (ANOVA, *p* < 0.01; Figure 1B), while, for the 10 °C decrease treatment, the transcripts were markedly elevated from 20 min to 180 min, reaching the highest (~five-fold greater than that at the beginning) (ANOVA, *p* < 0.01; Figure 1C).

In the experiment designed to compare the influences between moderate (5° C for each step) and drastic (10 °C for one step) temperature stress, while the transcript levels in the heating and cooling treatments exhibited similar patterns, the transcripts were not significantly elevated upon moderate temperature stress (± 5 °C) for 10 min (ANOVA, *p* > 0.05; Figure 2). Both the stepwise and one-step temperature shocks markedly elicited *StHsp90* transcription within 1 h (ANOVA, *p* < 0.01; Figure 2), while the transcript levels for both groups were statistically the same (ANOVA, *p* > 0.05; Figure 2).

### 2.4. Differential Transcription Profiles of StHsp90 at Different Stages of Growth and Life Cycle

To investigate the transcriptions of *StHsp90* at different stages of the growth and life cycles, vegetative cells at exponential and stationary growth stages, newly formed resting cysts, and resting cysts stored at 4 ± 1 °C in darkness for 20 and 30 days were compared for their transcriptions of *StHsp90*. While no detectable difference in *StHsp90* transcription was found among all the vegetative cell groups (including the exponential and stationary stages) and the newly formed resting cysts (ANOVA, *p* > 0.05; Figure 3), significantly higher expressions were detected in resting cysts that were maintained in dormancy for 20 and 30 days (ANOVA, *p* < 0.01; Figure 3). *StHsp90* expressions were not significantly different between that for 20 and 30 days (ANOVA, *p* > 0.05; Figure 3).

### 2.5. Identification and Phylogenetic Analysis of Nuclear Dinoflagellate Hsp90 Genes from a Dinoflagellate-Specific e-cDNA Library

The preliminary keywords search (heat shock protein, Hsp90; heat shock protein 90) returned 66 entries, which were manually queried in public databases for further accurate identification. After excluding sequences with very short coding region and those without the characteristic signatures of the 90-kDa Hsp gene family, a total of 29 sequences, with complete coding regions covering the characteristic domains of the Hsp90 family (Hsp90; pfam00183), were identified from the dinoflagellate-specific full-length e-cDNA library. All the 29 members included another domain of HATPase_c (histidine kinase-like ATPases; smart00387). These newly yielded sequences were deposited in GenBank under the accession numbers MZ825171-MZ825199. Among them, 21 entities contained the PRK05218 (Hsp90) superfamily domain. The other eight entities (MZ825171, MZ825178, MZ825179, MZ825185, MZ825186, MZ825187, MZ825190, and MZ825191) were predicted to belong to the superfamily of 83-kDa heat shock protein (Hsp83; PTZ00272).

With the aim to tentatively infer the possible taxa from which the 29 new *Hsp90* genes were derived, the BI distance tree formulated from the alignment of these *Hsp90* sequences together with the other 60 registered orthologous from dinoflagellates and other Alveolata species was evaluated by bootstraps. Sequences from the green alga *Ulva pertusa* and the higher plant model *Arabidopsis thaliana* were used as outgroups. The yielded tree showed explicitly that the Hsp90s newly captured in this study spread into two groups: 21 entities belonging to the Hsp90 superfamily clustered together as a branch and were at the basal position, and the other eight entities belonging to the Hsp83 superfamily and the *StHsp90* cloned from *Scrippsiella trochoidea* were grouped with other known Hsp90 sequences from the dinoflagellates (Figure 4). Among the Hsp83 group, three entities (MZ825171, MZ825179, and MZ825187) clustered together (with full node support) and nested within the thecate genus *Protoperidinium* with a high node support of 0.99, which was a sister clade to that of *Prorocentrum* (BI 0.87). A close affinity was revealed between *StHsp90* (MZ779085) and another Hsp90 sequence from *Scrippsiella trochoidea* with full node support (Figure 4). The sequence MZ825178 was a sister to those of the armored species *Kryptoperidinium foliaceum* and *Pentapharsodinium dalei* with a high node support of 0.99. The three entities, MZ825185, MZ825186, and MZ825191, clustered tightly (BI 0.99), showed a close affinity to MZ825190 (BI 1.00), and the four sequences together were sisters to that of the athecate species *Gymnodinium aureolum* and *Lepidodinium chlorophorum* (BI 0.98) (Figure 4).

## 3. Discussion

### 3.1. General Comments on the Newly Isolated Hsp90 Gene from Scrippsiella Trochoidea

In photosynthetic eukaryotes, the founding members that gave names to Hsp90s were divided into five types with different localizations of the cellular compartments: including nucleoplasm, chloroplast, mitochondria, endoplasmic reticulum, and cytoplasm [7,9]. The *Hsp90* gene from *S. trochoidea* in this study contains a cytosolic-specific motif MEEVD at the C-terminal region, suggesting it belongs to the cytoplasmic Hsp90s. This region has also been documented to serve as the major interaction site for co-chaperones of Hsp90s and facilitate Hsp90s functions in the folding and activation of substrate proteins [9]. In addition, the five typical Hsp90 signatures and the functional LXXLL sequence involved in binding to the nuclear receptor presented in *StHsp90* are in good accordance with the same special distributions detected in other well-known Hsp90 members [1,9]. The Hsp90 family is one of the most conserved molecular chaperones and has been identified in organisms that are evolutionarily distant [3,6,10]. The extremely well-conserved sequence signatures and motif of StHsp90 implies its conserved function(s) similar to those in other well-characterized organisms, although these functions have not been uncovered explicitly yet.

### 3.2. StHsp90 May Be Engaged in the Resistance to Both Heat and Cold Stresses

Abiotic stresses adversely affect the growth and productivity of plants and evoke a series of morphological, physiological, biochemical, and molecular changes [1,2]. The cellular response to stresses is a ubiquitous protective mechanism that triggers activation of the gene set involved in cell survival and/or cell death [3,4,8]. The Hsp90 members are crucial participants in the process, which shield cells from a wide variety of damaging stressors and are important for the recovery and survival of organisms [1,3,5]. As an essential component of the protective shock response, the fundamental role of Hsp90 members is to assist in the correct folding of nascent proteins and prevent the aggregation of stress-accumulated misfolded proteins, thus contributing to modulating the protein quality control and protein homeostasis [3,4,6,7,9,10]. In the present study, both higher (25 °C, 30 °C) and lower (15 °C, 10 °C, 5 °C) temperature shocks in comparison to the 20 °C control could stimulate *StHsp90* transcription significantly, implying that *StHsp90* production may function in the subsequent proteostasis restoration under both heat and cold stresses in *S. trochoidea.* The accumulated mRNA profiles of *StHsp90* observed in this study are parallel with a prior investigation also showing a transcriptional response of the *Hsp90* gene to both heat and cold shocks in another dinoflagellate, *Prorocentrum donghaiense* [44]. These results are also congruent with the thermally induced upregulation of *Hsp90* (or *Hsp90* homologs) in the dinoflagellates *P. minimum* [42] and *Symbiodinium* sp. [38]. It was noted that plants synthesize Hsps proportionally with the severity of the heat stress until the maximum level [3]. Our time series detections of *StHsp90* expressions exhibited a time-dependent manner upon the thermal and cold inducements that are supportive to the notion. The mRNA levels kept escalating within the 180-min exposures, indicating that the magnitude of the temperature stress continuously increased along with the exposure time and, thus, incurred a continuous transcription of *StHsp90*. Compared with low-temperature inducement, *StHsp90* seemed to be more sensitive to heat stress, since prominently elevated *St**Hsp90* transcription occurred within 10 min after the 10 °C increase exposure, and the maximum level at 180 min was up to 16-fold of that in the control group. The results described above together suggest an involvement of *StHsp90* in the emergent responses of *S. trochoidea* to temperature fluctuations, especially temperature escalation.

Interestingly, we found that both a moderate (5 °C for each step) and a drastic (10 °C for one step) temperature shock (increasing or decreasing) could elicit similar elevations of *StHsp90* transcription in one hour of exposure. In sharp contrast, our previous works observed that drastic stresses (one-step shocks ± 10 °C) stimulated significantly higher transcriptions of the *Hsp40* gene in *S. trochoidea* and the *Hsp70* gene in another blooming dinoflagellate, *Akashiwo sanguinea*, compared to mild stresses (5 °C for each step) [50,51]. It was proposed for terrestrial plants that thermotolerance is acquired through either a continuous, moderate heat shock experience to a preinduction procedure or a slow acclimation to increasingly severe heat shock [3]. It is also widely accepted that stress-inducible Hsps are strongly involved in the stress resistance of organisms [1,3]. Therefore, a parsimonious explanation for our results is that *StHsp90* may participate in the adaptation of *S. trochoidea* to severe temperature stress via cross-talks with other Hsp members (e.g., *Hsp70* and *Hsp40*) and/or function synergistically with other stress response mechanisms to maintain cellular homeostasis during dramatic temperature perturbations.

Temperature extremes is one of the most common abiotic stresses. The Hsp members are best known for their response to temperature variations [1]. The need for Hsp members is accelerated under a temperature shock that could potentially damage the cellular and molecular structures in the cells. Under normal physiological conditions, Hsps are constitutively and highly expressed, almost contribution to 1 to 2% of the total cellular proteins; while exposed to heat stress, the fraction increases to 4–6% [3]. Our recent findings in *S. trochoidea* revealed that several Hsp genes, *Hsp60*, *Hsp10*, *Hsp20*, and *Hsp40*, are modulated by temperature variations at the transcriptional level [47,50]. Together with the newly generated *StHsp90* in this study, these Hsp members are most probably involved in the adaptive response to both up and down temperature fluctuations of *S. trochoidea* and function together as part of the cell’s adaptive (and/or survival) strategy when it encounters temperature shocks. Our findings collectively contribute to a better understanding about the involvement of Hsp members in temperature resistance and pave the way for exploring their potential ecological implications in dinoflagellates.

### 3.3. Hsp90s Expressions Observed Both in Laboratory Culture and in the Cyst Assemblage in the Field Suggest an Involvement of Hsp90 Chaperones in the Dormancy of Resting Cysts of Dinoflagellates

In order to preliminarily probe the possible role of the *Hsp90* gene in relation to the life cycle transition of dinoflagellates, we attempted to characterize *StHsp90* mRNA abundance at different stages of growth and the life cycle in *S. trochoidea*. The qPCR results revealed that *StHsp90* mRNA expressions were statistically the same among vegetative cells at the exponential and stationary growth stages and newly formed resting cysts, suggesting that the need for *StHsp90* production was not accelerated during the transformation from vegetative cells into resting cysts. However, interestingly, we noted that prominently elevated *St**Hsp90* transcriptions in low temperatures and darkness-stored resting cysts compared to those in newly formed resting cysts and vegetative cells. The experimental design of those cysts maintained in dormancy at 4 ± 1 °C in darkness for different durations was aimed to simply imitate the natural conditions of overwintering marine sediments. Therefore, the above finding seemingly indicated that *StHsp90* was actively expressed during the resting stage of *Scrippsiella trochoidea*. To reveal what is happening in the natural environment, we used a unique dinoflagellate mRNA-specific spliced leader (DinoSL [32]) as the selective primer to generate a full-length e-cDNA library from a marine sediment sample and sequenced it using the third-generation sequencing technique. Therefore, all 29 Hsp90 genes with complete coding regions identified from the transcript pool of the marine sediments can be considered with certainty to be from dinoflagellate resting cysts and they were all actively expressed by the resting cysts during dormancy in the field. It is thus highly reasonable to infer that Hsp90s of dinoflagellates play essential roles during the dormancy of resting cysts in marine sediments. The phylogenetic reconstruction inferred from these 29 sequences and other known Hsp90 homologues from dinoflagellates and other Alveolata species suggests that these newly captured *Hsp90s* were not derived from any particular species or groups of dinoflagellates but are widely present in different taxonomic groups instead. These findings together point to a possibly universal presence of Hsp90s transcripts in the resting cysts of dinoflagellates buried in marine sediment, which is supported by the abundant transcripts obtained via qPCR for the cold-stored resting cysts of *Scrippsiella trochoidea* in the laboratory.

The Hsp90 chaperones distinguish themselves from other molecular chaperones in that most of their known substrates (clients) to date are regulatory proteins that reside in developmental and signaling pathways [4,9,10]. Due to their client spectrum, apart from stress responses, the most prominent influences of Hsp90 inhibition are detectable on signal transduction pathways and developmental programs [2,7]. Several interesting case studies for higher animals recorded that Hsp90 members were similarly upregulated during oogenesis or early embryogenesis, reflecting that they closely pertain to the assembly of centrosomes in early embryonic development [10,52,53]. Decreasing the functional Hsp90 level by genetic mutation or a Hsp90 inhibitor led to developmental abnormalities and morphological changes in *Drosophila* species [11]. The current knowledge on Hsp90s in terrestrial plants is much less advanced, and there are relatively fewer reports about their physiological functions, which mainly included their involvement in phenotypic plasticity, developmental stability, and the buffering of genetic variation (as reviewed in Reference [2]). In algae, based on the differential screening of the cDNA library, one *Hsp90* member was proposed to be involved in the differentiation of the female gametophytes in red alga *Griffithsia japonica* [54]. Another case work on dinoflagellate *Alexandrium catenella* found that one *Hsp90* was highly expressed at the protein level at an exponential growth stage, implying it might be related to maintain the proper order of the cell cycle progression of the species [36]. The characterization of Hsp90s in this study both in a laboratory-raised culture and the resting cyst assemblage of the field highlights possibly essential transcriptional machinery underpinning the dormancy maintenance of dinoflagellates, although it is still too early to speculate insightfully the exact roles it plays during the resting cysts’ dormancy. Based on the results from the other organisms reviewed above, a general perspective may be that Hsp90s likely serve in the adaptation or protection of dinoflagellates to natural settings and/or lie at the interface of physiological pathways via supporting (maturation and activation) signal transducers during the resting stage of dinoflagellates. Our results opened a window for the further functional characterization of Hsp90 together with other members of Hsps in dinoflagellates, an ecologically and biologically important lineage of protists.

## 4. Materials and Methods

### 4.1. Culture Establishment and Maintenance of Scrippsiella Trochoidea

The clonal culture of *S. trochoidea* strain IOCAS-St-1 obtained from the Marine Biological Culture Collection Centre, Institute of Oceanology, Chinese Academy of Sciences was originally established from the Yellow Sea of China and further confirmed species identity by partial LSU and SSU rRNA genes sequencing [46]. Cultures were maintained in sterile filtered natural seawater (salinity of 32 to 33) enriched with f/2-Si medium [55] and a penicillin–streptomycin solution (100×, Solarbio, Beijing, China) (final concentration of 2 to 3%) at 20 ± 1 °C, and illumination of 100-μmol photons m^−2^ s^−1^ (cool white fluorescent lights) with a 12 h:12 h light/dark photoperiod. The culturing conditions were also used in the following experiments unless otherwise indicated.

### 4.2. Cloning of the Full-Length StHsp90 cDNA

For gene cloning, ~10^6^ fresh vegetative cells from regular cultures were collected and used for total RNA extraction immediately. Total RNA was extracted according to Reference [50] and digested with Qiagen’s RNase-Free DNase Set following the manufacturer’s instructions. RNA concentration and quality were determined by NanoDrop^TM^ 1000 spectrophotometer (Thermo Fisher Scientific, Waltham, MA, USA).

For fragment amplification, the template of single-stranded cDNA was prepared from ~1 μg of total RNA with random primers and made with Reverse Transcriptase M-MLV (Takara, Tokyo, Japan). The cDNA fragment was amplified following the protocol described in Reference [51] with specific primers, P1 and P2 (Table 1), designing according to an *Hsp90*-like sequence found in the transcriptomic data of *S. trochoidea* (GenBank Accession No. SRP058465; [46]). The obtained cDNA fragments were then applied to design gene-specific primers (Table 1) for the 5′ and 3′ rapid amplification of cDNA ends (RACE) PCR to isolate the complete coding region via following the protocol described in Reference [51]. The templates used in RACEs were synthesized from ~1 μg of total RNA with the anchor primer (Table 1). The forward primer DinoSL (specific to dinoflagellates [32]) combined with reverse primers P3 and P4 (Table 1) were used for 5′ end amplification, while the forward primers, P5 and P6 (Table 1) paired with reverse primer GeneRacer3 (Invitrogen, Karlsruhe, Germany) were performed to generate the 3′ end. The final PCR products were then separated by electrophoresis in 1% agarose gels, purified using a Generay DNA gel extraction kit (Yantai, China), ligated into the *pEASY*-T1-cloning vector (TransGen Biotech, Beijing, China), transformed to Trans T1 *Escherichia coli* (TransGen Biotech, Beijing, China), and was sequenced at Sangon Biotech Company (Qingdao, China).

### 4.3. Analysis of StHsp90 Deduced Amino Acid Sequence

The potential protein-encoding segment in the yielded nucleotide sequence was searched via Open Reading Frame Finder [56]. A Simple Modular Architecture Research Tool (SMART) [57] was used to predict the conserved domains in the identified protein sequence, and the conserved domains were further confirmed by Pfam [58]. The predicted molecular weight and isoelectric point were calculated by the ProtParam tool [59]. The SOPMA program [60] was used to determine the secondary structure. The SignalP 4.0 Server was applied to check the signal peptide [61].

### 4.4. Transcriptional Profiles of StHsp90 in Responses to Temperature Stress and Alteration of Life Cycle in Laboratory Culture

#### 4.4.1. Samples Collection

##### Temperature Stress Exposures

To examine the transcription pattern of *StHsp90* in response to temperature variations, vegetative cells kept at routine maintenance conditions (20 ± 1 °C) were aliquoted into plates (6-well; Corning, Corning, NY, USA) and subjected to 3 temperature treatments, as described before [47,50,51], and briefed as follows: In the first one, the cells were immediately exposed to higher (30 and 25 °C) or lower temperatures (15 and 5 °C) for 60 min in incubators with the temperature set in advance. In the second time course exposure treatment, the cells were exposed to 10 and 30 °C, respectively, for 0, 3, 5, 10, 15, 20, 30, 60, 120, and 180 min in an incubator with preset temperatures; The last treatment was deigned to compare the influences between drastic and moderate temperature fluctuations. The drastic scenario was to put the cells directly to a temperature variation of 10 °C (exposure to 30 and 10 °C, respectively) for 60 min, whereas the moderate one was to first put the cells at temperature variations of 5 °C for 10 min and then subject them to a further 5 °C change for 60 min. The above 3 treatments were independently performed in triplicate. Vegetative cells kept at the routine maintenance temperature (20 ± 1 °C) were used as the control group. Roughly 2 ×10^5^ motile cells in each sample were harvested, immediately frozen in liquid nitrogen, and then kept at −80 °C.

##### Cells at Different Life Stages

In order to evaluate whether the transcriptional level of *StHsp90* was linked with the life stage alternation of *S. trochoidea*, vegetative cells and resting cysts were prepared by generally referring to previous works [47,48,49,50] and briefed as follows. For vegetative cells at different growth stages, triplicate cultures were inoculated in ~300-mL medium in 500-mL flasks with an initial density of ~2000 cells mL^−1^ to obtain the growth curve of *S. trochoidea*. Three-milliliter culture samples were taken every day and fixed with Lugol solution. Cell density was determined via daily counting using a plankton counting chamber with an inverted light microscope (IX73, Olympus, Tokyo, Japan). The inoculation day was recorded as Day 0. Based on the growth curve (see Appendix A for more details), the samples were collected on Days 5 and 9 and Days 12 and 15 for cells at the exponential and stationary growth stages, respectively.

Resting cyst samples were prepared by referring to References [47,48,49,50] and briefed below. Resting cysts were produced and harvested from cultures grown in 6-well culture plates (Corning, Corning, NY, USA) and kept at routine culturing conditions, except for that the medium was made with artificial seawater supplemented with all nutrients of the abovementioned recipe but nitrogen and phosphorus. The vegetative cells and immature and mature cysts differed markedly in morphology via light microscopy [46,48,50]. The cultures were checked for cyst formation every 2 days under an Olympus IX73 inverted microscope. Resting cysts were obtained from the cultures that were inoculated for about 30 days. The harvested cysts were cleaned with fresh sterile filtered seawater several times until no vegetative cell or planozygote (motile zygote) were observed in the samples by checking under a microscope. The obtained resting cysts (in 6-well culture plates; Corning) were then kept at 4 ± 1 °C in darkness for 20 and 30 days, respectively. A total of 3 cyst samples, newly formed, maintained at 4 °C in darkness for 20 and 30 days, respectively, were included in the following qPCR analysis. For each above-mentioned sample, approximately 2 ×10^5^ vegetative cells or resting cysts with 3 biological replicates were prepared, frozen immediately in liquid nitrogen, and then kept at −80 °C.

#### 4.4.2. Real-Time Quantitative PCR (qPCR)

The qPCR reactions were conducted with the SYBR^®^
*Premix Ex Taq*^TM^ (TaKaRa, Tokyo, Japan) on the Bio-Rad CFX96 real-time PCR detection system (Bio-Rad Laboratories, Hercules, CA, USA). For each sample, the first-stranded cDNA synthesized from ~60-ng total RNA with random primers was used as the template in the following detection. The protocol and cycling conditions generally followed those described in References [49,50]. A specific primer set, P7 and P8 (Table 1), was designed based on the full-length cDNA sequence of *StHsp90* generated above to amplify the 129-bp product. The reference gene combination of *MDH* (malate dehydrogenase), *UBC* (ubiquitin conjugating enzyme), and *LBP* (luciferin-binding protein) was used as the internal control in the qPCR detection of *StHsp90* transcriptions at different life stages [46]. The combination of *UBQ* (ubiquitin), *MDH*, and *UBC* was applied in qPCR analyses of vegetative cells subjected to temperature stresses [49]. All the amplicons were sequenced to confirm the correct amplifications. All the reactions were performed in biological triplicates. The melting curve analysis was conducted to demonstrate the specificity of each PCR product. For each primer pair, the relative standard curve [62] and qPCR efficiency (*E*) [63] were calculated as previously described in Reference [47]. Relative quantitative values were calculated with the 2^−∆∆Ct^ relative quantification method by using qBasePlus software [64,65]. The relative expression levels among the different groups were analyzed using one-way analysis of variance (ANOVA). Significance was inferred when *p* ≤ 0.05. Statistical analyses were conducted with SPSS 20.0.

### 4.5. Construction of Dinoflagellate-Specific Environmental cDNA (e-cDNA) Library from Marine Sediment

#### 4.5.1. Sediment Collection and Resting Cysts Separation

Sediment sample was taken from Jiaozhou Bay, Qingdao, Shandong Province, China (36.159° N, 120.229° E) on 24 May 2016. The upper 10 cm of sediment was collected by using a grab sampler, transferred to sterile plastic bags, and kept in an ice-contained cooler to maintain the darkness and low temperature. After being transported to the laboratory, the sediment sample was immediately used for resting cyst separation. More than 200 g (wet weight) of surface sediment (at a layer of 3-6 cm deep) was used following the standard protocols described in Reference [66] by using sodium polytungstate (SPT). The harvested resting cysts were washed with fresh sterile filtered seawater several times, concentrated by centrifugation, and immediately used for the subsequent RNA extraction.

#### 4.5.2. Construction of e-cDNA Library and PacBio Iso-Seq Sequencing

Total RNA was extracted from all the harvested resting cysts, as described in Section 4.2. The single-stranded cDNAs were synthesized from ~1-μg total RNA and made with a SMARTer PCR cDNA Synthesis Kit (Clontech, USA) by using the anchor primer (Table 1). The resultant cDNAs were then purified with a Zymo DNA Clean and Concentrator Kit (Zymo Research, Orange, USA) and used as templates in the PCR amplifications. To minimize the PCR bias, PrimeSTAR GXL DNA High Fidelity Polymerase (Takara, Tokyo, Japan) was used. PCR was performed with the specific primers DinoSL (specific to dinoflagellates [32]) and Racer3 [49] to amplify the dinoflagellate-specific cDNAs under the program: 94 °C for 3 min, followed by 95 °C 15 s and 68 °C 3 min 30 s for 5 cycles; 95 °C 15 s, 62 °C 30 s, and 72 °C 3 min for 15 cycles; and a final extension at 72 °C for 10 min. The sequencing library was constructed generally according to the Isoform Sequencing (Iso-Seq) protocol and the BluePippin Size Selection System protocol as described by Pacific Biosciences (PN 100-092-800-03). Size fractionation and selection were performed using the BluePippin with the following bins: 0.5-1.8 kb and > 1.8 kb. The DinoSL-based e-cDNA library underwent single-molecule real-time (SMRT) sequencing using 6 SMRT cells on the PacBio Sequel platform. Sequencing was performed by BGI Genomics Co., Ltd. (Shenzhan, China).

#### 4.5.3. Data Processing

The raw SMRT sequencing reads were subjected to SMRT Analysis Server v2.3.0 supported by PacBio to filter out low-quality polymerase reads (read length <50 bp and read score <0.75). The CCSs (circular consensus sequences) were filtered from the subreads with the full-pass threshold set to ≥0 and the predicted unique accuracy set to ≥ 0.75. Based on examining for cDNA primers and a poly (A) signal, the CCS reads were classified into full-length non-chimeric (FLNC), non-full-length (nFL), chimeras, and short reads. Short reads less than 500 bp were discarded. The CCSs with all three elements (5′-cDNA primer, a 3′-cDNA primer, and a poly (A) tail) and not containing any additional copies of the adapter sequence within the DNA fragment were selected and classified as FLNC (full-length non-chimeric) sequences. Then, 5′- and 3′-cDNA primers and poly (A) tail were removed from FLNCs according to the PacBio recommended procedure. The obtained FLNC reads were subsequently clustered by Iterative Clustering for Error Correction (ICE) algorithm to generate the cluster consensus transcripts. Then, the nFL reads were used to polish the above cluster consensus transcripts to yield the full-length polished high-quality consensus sequences (accuracy ≥99%). The final full-length cDNA sequences were yielded by removing the redundant sequences with software CD-HIT using a threshold of 0.98 identities [67].

### 4.6. Identification of Hsp90 Genes from Dinoflagellate-Specific e-cDNA Library and Phylogenetic Analysis

The sequence homology and predicted functions of the generated SMRT genes were annotated using BLAST2GO under the translated nucleotide BLAST (BLASTX) algorithm. Then, the transcript pool was searched by keywords heat shock protein, Hsp90, and heat shock protein 90 to collect potential Hsp90 candidates. The retrieved entries were further confirmed by conducting BLASTX searches against public databases, including the NCBI nonredundant protein database (Nr) database, NCBI nonredundant nucleotide database (Nt) database, SwissProt database, InterPro database, with a 10^−6^ E-value cutoff. The potential protein encoding segments in the identified nucleotide sequences were searched via Open Reading Frame Finder [56]. The Simple Modular Architecture Research Tool (SMART) was used to predict the conserved domains in the identified protein sequences [57].

The 29 newly identified Hsp90 sequences from the dinoflagellate-specific full-length e-cDNA library, together with the new generated *StHsp90* and the other 60 known Hsp90 homologs from dinoflagellates and other alveolata species (Appendix A) downloaded from the GenBank database, were aligned by using MAFFT v.7 with the G-INS-i algorithm [68]. The alignment was further refined manually with BioEdit 7.0.9.1 [69]. A Bayesian inference (BI) analysis was performed with MrBayes 3.2.6 [70]. The posterior probability was estimated using four chains running 1,000,000 generations, and trees were sampled every 100 generations. The first 25% sampled trees were discarded as burned in prior to generating a 50% majority rule consensus tree. The posterior probabilities for all the branches were calculated using a majority rule consensus approach. The phylogenetic trees were visualized using FigTree v1.4.4.

## 5. Conclusions

We isolated the full-length cDNA of a Hsp90 gene (*StHsp90*) via RACE from the cosmopolitan HAB species *Scrippsiella trochoidea* and tracked its transcriptions in response to varied scenarios via real-time qPCR. The qPCR results together suggested that *StHsp90* is probably related to the resistance of dinoflagellates to both heat and cold stresses. Furthermore, we observed prominently elevated *St**Hsp90* transcriptions in the cysts that were stored at cold and dark conditions compared to those in newly formed resting cysts and vegetative cells, indicating an active expression of *StHsp90* during the dormant stage of dinoflagellates. Then, we identified 29 entries of *Hsp90*-encoding genes with complete coding regions from a dinoflagellate-specific environmental cDNA library generated from marine sediment assemblages, suggesting the active transcription of these *Hsp90* genes in the marine sediment-buried resting cysts assemblage of dinoflagellates, which was fully supported by the qPCR results for the cold-stored resting cysts of *Scrippsiella trochoidea* in the laboratory. The evaluation of *Hsp90* expressions both in the laboratory-raised culture and field cysts assemblage collectively highlighted a possible involvement of Hsp90 chaperones in the resting stage persistence of this ecologically important lineage of protists.

## Figures and Tables

**Figure 1 ijms-22-11054-f001:**
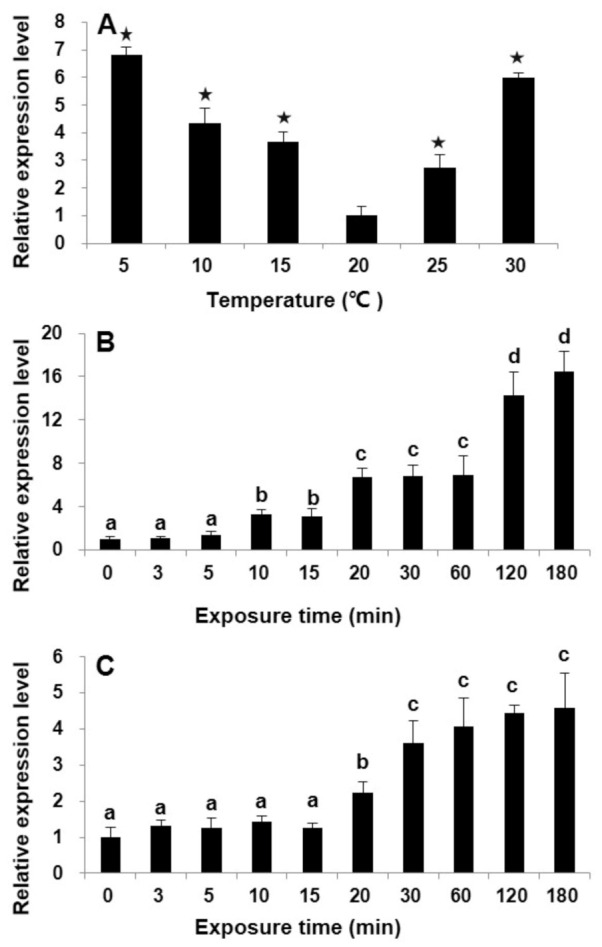
The transcription levels of *StHsp90* in cells under different shock temperatures (**A**) and during a 180-min time course with a 30 °C treatment (**B**) and 10 °C exposure (**C**). Each bar represents the mean of three biological replicates, and error bars signify the standard deviation of those means. Asterisk indicates significant increases at the transcription level as compared with the 20 °C group (*p* < 0.05) (**A**). Same letter above the bar denotes no significant difference (*p* > 0.05) in abundance (**B**,**C**).

**Figure 2 ijms-22-11054-f002:**
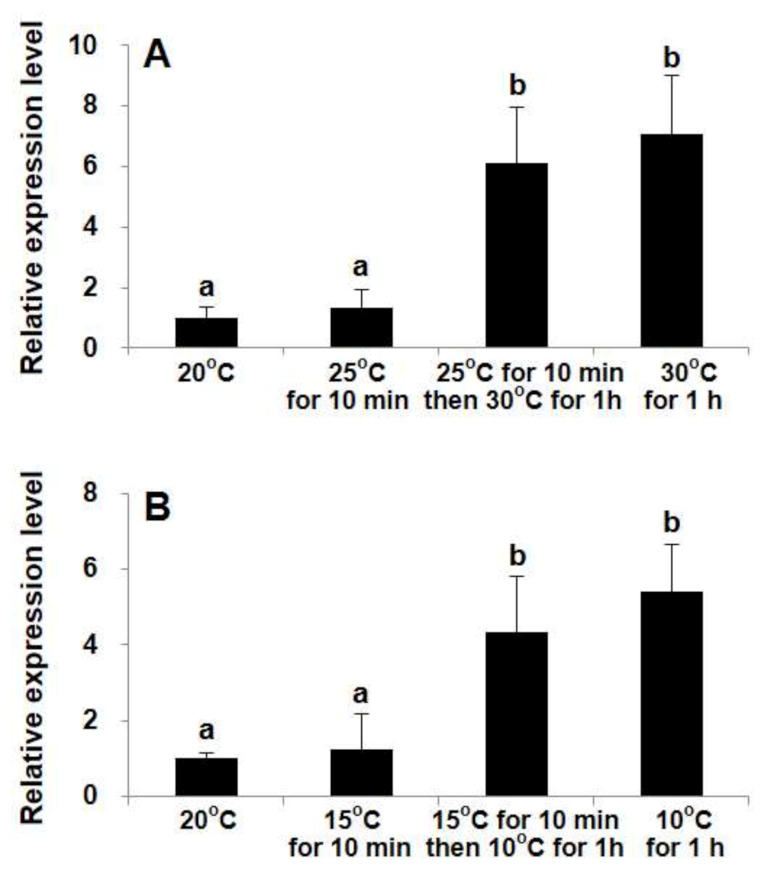
The transcription levels of *StHsp90* in cells exposed to one-step or stepwise temperature shock (+10 °C (**A**) and −10 °C (**B**)). One-step shock: put the cells directly in a temperature variation of 10 °C for 60 min. Stepwise shock: first, put the cells in a temperature variation of 5 °C for 10 min and then subject them to a further 5 °C change for 60 min. The 20 °C maintained cultures are used as the control. The relative expression values are shown as the mean fold changes compared to the control, with the error bars depicting the standard deviations (*n* = 3). Same letter above the bars denotes no significant difference (*p* > 0.05) in abundance.

**Figure 3 ijms-22-11054-f003:**
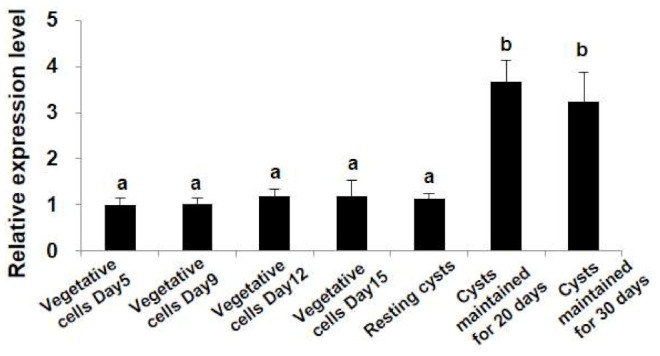
The transcription levels of *StHsp90* in cells at different life stages (vegetative cells at the exponential and stationary growth stages, newly formed resting cysts, and resting cysts in a forced dormancy at a lower temperature and in darkness for 20 and 30 days, respectively). The relative expression values are shown as the mean fold changes compared to the control (vegetative cells collected at Day 5), with the error bars depicting the standard deviations (*n* = 3). Significant differences are indicated with different letters at *p* < 0.05.

**Figure 4 ijms-22-11054-f004:**
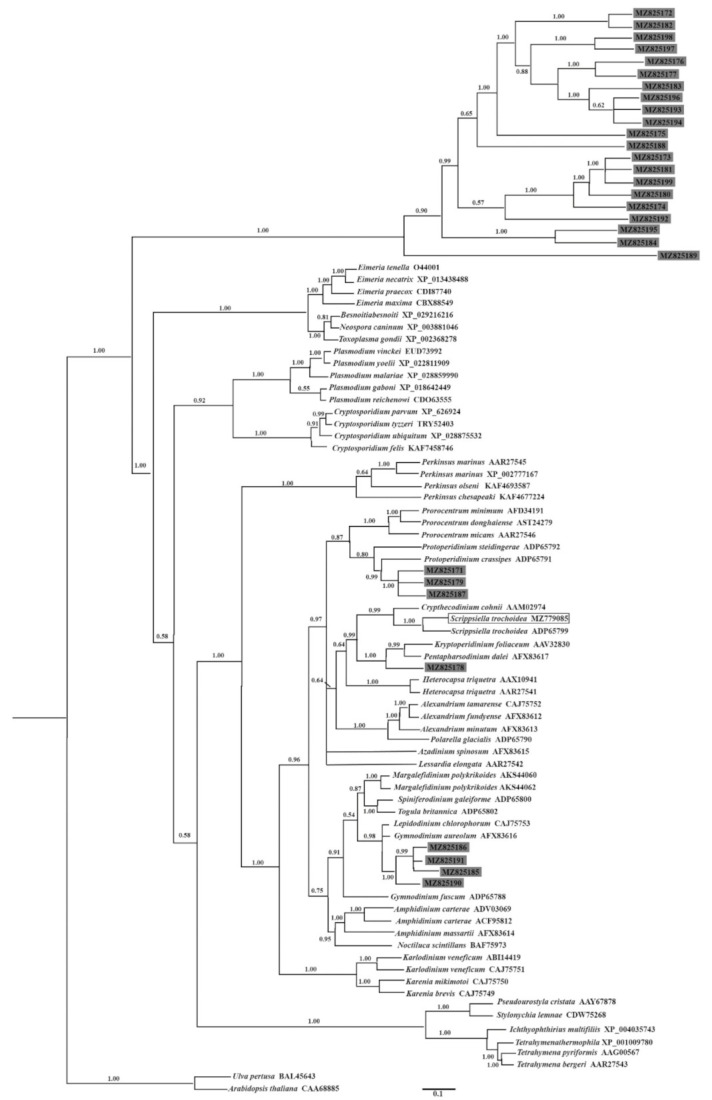
The Bayesian inference (BI) distance tree formulated from the alignment of the full amino acid sequences of the complete coding regions of the 90 Hsp90 sequences. Numbers at the nodes represent the BI posterior probabilities. For the 60 entries downloaded from the GenBank database, the accession number of each sequence is noted following the species name (see Appendix A for more details). The newly generated *Hsp90* gene from *Scrippsiella trochoidea* in the current study is boxed. The 29 newly identified *Hsp90* sequences from the e-cDNA library are present as the GenBank accession number highlighted in the gray background.

**Table 1 ijms-22-11054-t001:** List of primers in this study.

Primer Name	Nucleotide Sequences (5′→3′)	Remarks
P1	TCTTCATTATGGACGATTGC	fragment cloning
P2	GATGGAGTGCTTCGGATT	fragment cloning
P3	GCACTTGCCAAACTGCTCGTAACATT	5′ RACE
P4	CACGACGCCCTTGACCATGTTGA	5′ RACE
P5	ATGGCTGACTCCCCTTGCGTGCTC	3′ RACE
P6	GAGGTGAATCCGAAGCACTCCATC	3′ RACE
P7	GCATTCGGAATTTATTGGC	qPCR
P8	ATCTTCGGCTCGTCACCC	qPCR
anchor primer	GCTGTCAACGATACGCTACGTAACGGCAT GACAGTGT (18)	cDNA synthesis

## Data Availability

The data is contained within the article or the Appendix A.

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
