# Peer review of "Expression Patterns of the Heat Shock Protein 90 (Hsp90) Gene Suggest Its Possible Involvement in Maintaining the Dormancy of Dinoflagellate Resting Cysts"

_ijms, 2021, doi:10.3390/ijms222011054_

Round 1

Reviewer 1 Report

Minor remarks:

Introduction

Abassi et al., 2020 (Cell Stress and Chaperones) – can add

Results

Figures 1 and 2 – can be transferred to the application.

Space before “˚C” – make it the same throughout the text

No space before “%” - make it the same throughout the text

Table 1 is referenced on line 423 and should appear immediately at the top of the next page.

Minor corrections:

Line 662 – should be “BMC Genomics” instead of “BMC genomics”

Line 679 - should be “Chaperones” instead of “Chaperone.” – “.” Extra point also

Line 683 – delete “,” after “Genomics”

Line 703 – “50” should be italic

Line 734 – delete “,” after “…matics”

Line 631 – should be space between “Kim,” and “D.I.”

Line 708 - should be “O.-K.” instead of “O-K”

Line 716 – Who is hiding under “et al.”?

Conclusions – need to add.

Reviewer 2 Report

I have read the MS with great interest. The manuscript brings a new and original view of the ecology and ecophysiology of Gymnophyceae.  It demonstrates the role of heat shock proteins in the protection of Gymnophyceae against abiotic environmental stresses. The manuscript is showing experimental laboratory results which are combined with analyses of resting cysts in marine sediment. I am just recommending to shorter title and abstract. Also, English could be improved. I feel in the MS there are so long sentences. I support the MS should be published.
